# Evolution and Influencing Factors of Social-Ecological System Vulnerability in the Wuling Mountains Area

**DOI:** 10.3390/ijerph191811688

**Published:** 2022-09-16

**Authors:** Huiqin Li, Yujie Hui, Jingyan Pan

**Affiliations:** School of Economics and Management, China University of Geosciences, Wuhan 430074, China

**Keywords:** society-ecosystem, vulnerability, spatial-temporal evolution, Wuling Mountains area

## Abstract

With the wide spread of the concept of sustainable tourism in various countries and regions, the research on tourism poverty alleviation is paying increasingly closer attention to the sustainability of the poverty reduction effect of tourism, and the social-ecosystem theory of tourist destinations has been widely applied in the sustainable development of tourism in backward mountainous areas. However, existing studies lack the dynamic evaluation of social-ecosystem vulnerability in places of tourism, and are devoid of large sample data. This paper aims to analyze the law of spatial and temporal evolution of the social-ecosystem vulnerability in China’s Wuling Mountains area, and to help solve the obstacles and difficulties of realizing the effective connection between poverty alleviation and rural revitalization. The set analysis method (SPA) was used to evaluate the vulnerability and the obstacle degree model was utilized to identify the vulnerability barrier factors. Forty-two national key poverty-alleviation counties in the Wuling Mountains area were selected as the research objects to analyze the spatial and temporal evolution characteristics of social-ecosystem vulnerability based on the valuation model of “Social-Economic-Ecological (S-E-E) model” and the “Vulnerability-Scoping-Diagram (V-S-D) model”. In this paper, we clarified the two types of changes in social-ecosystem vulnerability in the Wuling Mountains area, and analyzed the spatial differences of vulnerability from the perspective of subsystems and counties. In terms of the results of this study, from 2010 to 2019, the overall vulnerability of social ecosystems showed a trend of “slow-rise and steady-decline”, with the vulnerability index peaking in 2014 and declining year by year thereafter. Spatially, the overall vulnerability is smaller in the north than in the south; and social-ecosystem vulnerability is the result of the interaction between system exposure-sensitivity and system adaptive capacity. Based on the change in vulnerability of different subsystems and different counties, and the barrier factors it faced, we make targeted suggestions to help the region to reduce its social-ecosystem vulnerability.

## 1. Introduction

With the continuous expansion of the intensity and scope of human activities, the human social-economic system has produced a complex relationship with natural ecosystems, and the study of ecosystems has expanded from the natural ecosystem to the social ecosystem of human daily production and life. Cumming suggested that social ecosystems are complex adaptive systems connecting human beings and nature [1]. The system analysis framework consists of such four subsystems: resource system, resource unit, management system and user, as well as their interrelationships [2]. The adaptive cycle theory was used by the resilience alliance to explain the social-ecosystem development, evolution process and mechanism. With the help of three dimensions of “potential-connectivity-toughness”, they linked the ecosystem evolution development, protection, release, update into a closed dynamic cycle. The theory is used to explain the complex ecosystem response to interference and change feedback dynamic mechanism [3]. Social-ecosystem is a highly potential analytical framework for sustainable development [4]. Among them, resilience, adaptability and vulnerability are the key attributes of the system. Vulnerability, a characteristic attribute of social ecosystem, was originally noticed by the academia in 1960, and later introduced into the field of geoscience by the scholar Timmerman [5]. Vulnerability refers to the degree of adverse effects when it cannot deal with disturbance caused by the external environment [6]. At present, it has become a hot issue in studying regional human–land relationship, and also is an important theoretical method for studying regional society [7], economy [8], environment [9] and the sustainable development of population and resources [10]. The social ecosystem explicitly recognizes the connection and feedback between human and natural systems [11]. In addition, it has broken through the previous cognitive limitations of separating human social-economic system from natural-ecological system [12], so it is widely used in sustainable governance. Its focus is on interdependent associations between social and environmental change, and how these interdependent connections affect the achievement of sustainable goals of different systems, levels and scales [13]. The related studies mostly focus on the three subsystems: economy [14], society [15] and ecology [16].

Due to its remarkable economic benefits and extensive driving effect, tourism has become an effective way to get rid of poverty in many remote and underdeveloped areas, which has been widely agreed upon by the industry and academic circle [17]. Guiding the local poor people to participate in tourism has become a preferable method to effectively alleviate poverty and prevent the return to poverty [18]. Investments can help improve the standards of living through job creation, skills and technology development and wealth distribution. Nevertheless, irresponsible business practices will not only erode the business environment, but they may also lead to economic losses, environmental degradation and even personal injury and loss of life [19]. The disorderly development of the tourism industry strongly disturbs the social ecosystem (SES) in poorly developed areas and increases the load of ecological environment [20], and its impact on economic development and unequal wealth distribution is also controversial [21,22]. As the concept of sustainable tourism has spread widely in various countries and regions [23], the research on tourism poverty alleviation is attaching much more significance to its sustainability. Furthermore, treating tourism as a whole and system can further stimulate the potential for sustainable poverty reduction [24].

Social-ecosystem is a complex system where the social subsystem (human) and ecological subsystem (nature) interact [25]. Disturbed by its own properties and its external environment, it presents complex features such as unpredictability, self-organization, multi-stability and threshold effects [26]. Compared with the general social ecosystem, the social ecosystem in areas of tourism poverty-alleviation has a highly close exchange relationship with the external environment (such as human flow, capital flow and information flow) due to the needs of tourism development, so it shows a rather open state [27,28]. It is faced with the dual influence of external human activities and system internal vulnerability. With the expansion of tourism scale, the social-ecological pressure felt by areas of tourism poverty-alleviation is increasing [29]. With the continuous growth of tourist demands and the continuous acceleration of the development of tourist areas, the alienation, imbalance and conflict in the human–land relationship are also increasingly intensified [30]. Environmental degradation, in turn, can pose a serious threat to tourism activities [3,26].

Tourist destination is the basis of tourism development, but also a relatively complete social-ecological system [31]. Farrell regarded the tourist destination as a multiple and complex adaptation system [32]. Petrosillo discussed the connotation of social-ecological tourism-based systems, and put forward management measures to reduce system vulnerability to different types of tourism places [33]. Social-ecosystem theory framework and analysis method can effectively address the changes in human–land relationship caused by tourism development; it also reveals the relationship between the development process and evolution of tourism social ecosystem. The theory has thus been widely applied to the sustainable development of tourism in poor mountainous areas.

However, Calgaro and Cole stated that research on the vulnerability of tourist destinations from the perspective of human–land relationship is scarce [34,35]. The existing studies lack the dynamic evaluation of social-ecosystem vulnerability, and the evaluation method of social-ecosystem vulnerability needs to be improved. At the same time, most of the existing studies are based on small sample datasets or cases such as counties, cities or provinces, devoid of large sample datasets. Therefore, the study in this paper uses a large sample of data with a long time series across regions to make up for the literature shortage, and screens the appropriate evaluation indicators to improve the defects of the social-ecosystem vulnerability evaluation method in China’s tourist areas.

The Wuling Mountains area runs through the three Provinces of Hunan, Hubei and Guizhou, plus the City of Chongqing in China. Its natural ecological environment is fragile, with backward social and economic development, and a large population of ethnic minorities. It used to be a contiguous poverty-stricken area integrating ethnic minorities, old revolutionary bases and remote mountainous areas, with prominent social-ecosystem vulnerability. Tourism poverty alleviation has promoted income increases for farmers and social stability in the Wuling Mountains area. In addition, the local infrastructure, health conditions, public security conditions and cultural literacy for local residents have all been significantly improved. However, the development of tourism poverty alleviation projects has occupied cultivated land and forest land, resulting in land shortage, and the ecological environment of some tourism land has been significantly degraded [36]. Meanwhile, rising prices and living costs, the widening gap between the rich and the poor, along with the deprived rights of the poor have exerted a negative impact on the social and economic system of the tourist area [37]. In the “post-poverty alleviation era”, it is of great significance to clarify the evolution characteristics of social-ecosystem vulnerability of tourist sites in this area in order to better facilitate tourism in the process of rural revitalization.

This study takes 42 state-level key poverty-stricken counties in The Wuling Mountains area as the research object. We started from the change in tourism poverty alleviation to the social-ecosystem vulnerability of the area. Then, we combined the “Vulnerability-Scoping-Diagram” framework and the “Social-Economic-Ecological” model to measure the social-ecosystem vulnerability from 2010 to 2019. This study aims to analyze the vulnerability characteristics of economic, social and ecological subsystems from the dimensions of time and space, and objectively evaluated whether tourism poverty alleviation improves the social-ecosystem vulnerability of the destination. At the same time, the obstacle degree model is used to identify the obstacle factors and influence mechanisms that affect the reduction in social-ecosystem vulnerability in the research area. We further proposed optimized control strategies. We hope to supplement the relevant scientific research on the long timing dynamic evaluation of social-ecosystem vulnerability in areas of tourism in the process of social-ecosystem vulnerability research in the Wuling Mountains area. We also wish to improve the evaluation method of social-ecosystem vulnerability in regions of tourism and to address the lack of large sample datasets in the existing literature. It is also expected to facilitate the Wuling Mountains Area to optimize its development policies, implement classified policies and adapt measures in line with local conditions, a combined effort to provide policy reference for helping the high-quality development of tourism, promoting rural revitalization and realizing China’s aim of “common prosperity.”

## 2. Materials and Methods

### 2.1. Study Area and Data Sources

The Regional Development and Poverty Alleviation Plan of The Wuling Mountains Area (2011–2020) divides the Wuling Mountains area into 42 key counties for national poverty alleviation and development work, 13 provincial-level key counties and 11,303 poverty-stricken villages. In this paper, 42 state-level poor counties were selected as study areas (Figure 1). Forty-two state-level poverty-stricken counties span Hunan, Hubei, Chongqing and Guizhou provinces, covering an area of about 97,400 square kms, accounting for 56.7% of the Wuling Mountains area. Among them, there are 10 counties and districts in Hubei province, 7 counties in Chongqing city, 15 counties in Hunan province and 10 counties in Guizhou Province. In 2019, the study area had a population of 21,236,200 people, including about 45% from the minority groups such as Tujia, Miao and Dong.

Since the implementation of the tourism poverty alleviation strategy, tourism revenue had increased from 29.009 billion yuan in 2010 to 310.49 billion yuan in 2019, with an average annual growth rate of 107.81%. The study area is rich in species and endowed with fine ecological environments. However, due to the frequent occurrence of heavy rain, flood, drought, geological disasters, soil erosion and rocky desertification caused by climate and natural environment, the carrying capacity of the ecological environment in this area is on the decline with a prominent ecological vulnerability.

In view of the availability of data, six time points in 2010, 2012, 2014, 2016, 2018 and 2020 were selected, and the 2020 data were replaced with 2019 data to exclude the impact of the COVID-19 outbreak. Natural geographic data were retrieved from the Data Center for Resources and Environmental Sciences, Chinese Academy of Sciences. The social and economic data were derived from the statistical yearbooks of various provinces, cities and autonomous regions in the corresponding years, as well as national economic and social development bulletin, government work report, *China County Statistical Yearbook* and *China Regional Economic Statistical Yearbook* [38,39]. Moreover, the environment related indicators are from environmental quality bulletins, while counties and scenic data are from provincial departments of culture and tourism.

### 2.2. Construction of the Evaluation Index System

The evaluation index of social-ecosystem vulnerability includes three subsystems: economic, social and ecological systems. Suggested by Pandey R., the V-S-D vulnerability evaluation model is suitable to solve the regional vulnerability evaluation caused by regional multi-scale and multi-factor perturbation [40]. By referring to the existing research results [41], considering the facts of the Wuling Mountains area and the opinions of relevant experts, we constructed the social-ecosystem vulnerability evaluation framework of the Wuling Mountains area, utilizing the SEE model and VSD framework. Similar to most studies, we worked mainly from social, economic and ecological aspects. However, according to the differences of social ecosystem types, the exposure, sensitivity and acclimatization of each system will be different, and the selection of evaluation indicators will also vary accordingly. Pandey R. used such indices as local farmers’ perception of climate change, experienced adaptation constraints, and family economic and social status, property and social relations to measures the vulnerability of social ecosystems in the Central Himalaya, Nepal [40]. Ren, G. adopted such indicators as total value of farm output, ratio of settled and farmland, number of agricultural workers, deserted farmland area and land degradation to characterize the state results and comprehensive coping abilities of the rural social-ecosystem vulnerability in the study of rural social-ecosystem vulnerability in Qingpu District of Shanghai [42]. The internal and external pressure of tourism places comes from the resource consumption, ecological destruction and external dependence brought about by the development of tourism industry. The material and energy exchange are affected by multiple factors, and its social and ecological environment have certain uniqueness. Therefore, the indicators selected in this study are closely related to tourism, such as total tourism revenue (D_1_), tourism revenue growth rate (D_2_), tourism economic density (D_3_), proportion of total tourism revenue in GDP (D_4_), tourist quantity (D_9_), tourist number growth rate (D_10_) and tourist population density (D_12_). These indicators can very well reflect the impact of tourism on the local economic and social systems. At the same time, the environmental protection expenditure (D_20_), the forest coverage rate (D_21_) and the artificial afforestation area (D_22_) are used to reflect the particularity of the research area as a mountainous area.

Meanwhile, the index weights were determined according to the entropy method, as shown in Table 1. Entropy value method is determined according to the degree of variation of the index value, which is an objective empowerment method avoiding the bias caused by human factors. Compared with those subjective assignments, its accuracy is more objective and can better explain the results. Compared with principal component analysis and factor analysis, it avoids the defect that only the weight of each factor is available with the specific weight of each analysis item unavailable. At the same time, it also avoids the problem of losing a small amount of information after the index dimension reduction. Due to the inconsistency of the index scale (units), the data of different indicators will vary greatly, which will affect the calculation results. To eliminate the effect of dimensionality, the data have been dimensionless.
(1)$$\begin{array}{c} \mathrm{MMS}=\frac{X-\mathrm{Min}}{\mathrm{Max}-\mathrm{Min}}\\ \mathrm{NMMS}=\frac{\mathrm{Max}-X}{\mathrm{Max}-\mathrm{Min}} \end{array}$$
where, X is the data for a certain index, Max represents the maximum value and Min represents the minimum value. MMS indicates results after the nondimensionization of the positive indicators.

### 2.3. Evaluation Method of Social Vulnerability to Ecosystem

The tourism social ecosystem is a complex system with a large number of deterministic and non-deterministic factors. This study introduces a set pair analysis method (SPA) to calculate the social-ecosystem vulnerability of the study areas. Set pair analysis is to treat the interconnected set E and set U as the set pair H. Under the condition of specific problem Q, it treats the certainty of the set pair as the opposite and unity, and uncertainty as difference, thus obtaining the N (number of) features of the set pair H, as well as the shared features S and opposite features P of set E and set U. The uncertainty features are F = N-S-P, and then comes the connection degree (μ) of the two sets:(2)$$\mu =\frac{S}{F}+\frac{F}{N}i+\frac{P}{N}j=a+\mathrm{bi}+\mathrm{cj}\;$$

The system vulnerability of various counties and districts in the Wuling Mountains area is established as Q = {A, B, W, D}, and evaluation scheme E = {e_1_, e_2_, e_3_,…, e_m_}. Each evaluation scheme has n indicators G = {g_1_, g_2_, g_3_,…, g_n_}. The corresponding metric weight is W = {w_1_, w_2_, w_3_,…, w_n_}, the value of the evaluation index is the one of d_kp_ (k = 1, 2, 3,…, m; p = (1, 2, 3,…, n), and the matrix of the problem Q is:(3)$$D=\left(\begin{array}{ccc} d_{11} & \cdots & d_{1n}\\ \vdots & \ddots & \vdots \\ d_{m1} & \cdots & d_{\mathrm{mn}} \end{array}\right)$$

The evaluation indices of the social-ecosystem vulnerability of the tourist area in the Wuling Mountains area were compared and analyzed to determine the optimal set and the worst set of the scheme, which are respectively U = {u_1_, u_2_, u_3_,…, u_m_} and V = {v_1_, v_2_, v_3_,…, v_n_}. According to the collection {v_p_, u_p_}, the same degree a_kp_ and opposition degree c_kp_ of d_kp_ in matrix D are as follows:

When d_kp_ is positively correlated with the evaluation results, there are:(4)$$\{\begin{array}{c} a_{\mathrm{kp}}=\frac{d_{\mathrm{kp}}}{u_{p}+v_{p}}\\ c_{\mathrm{kp}}=\frac{u_{p}v_{p}}{d_{\mathrm{kp}}\left(u_{p}+v_{p}\right)} \end{array}$$

When d_kp_ is inversely correlated with the evaluation results, there are:(5)$$\{\begin{array}{c} a_{\mathrm{kp}}=\frac{u_{p}v_{p}}{d_{\mathrm{kp}}\left(u_{p}+v_{p}\right)}\\ c_{\mathrm{kp}}=\frac{d_{\mathrm{kp}}}{u_{p}+v_{p}} \end{array}\;$$

Set-pair connection degree (μ) is:(6)$$\{\begin{array}{c} \mu_{\left(E_{k},U\right)=a_{k}+b_{\mathrm{ki}}+c_{\mathrm{ki}}}\\ a_{k}=\sum w_{p}a_{\mathrm{kp}}\\ c_{k}=\sum w_{p}c_{\mathrm{kp}} \end{array}$$

Scheme E_k_ and the sticking degree r_k_ of the optimal scheme can be defined as:(7)$$r_{k}=\frac{a_{k}}{a_{k}+c_{k}}$$
where, the larger the value of r_k_, the closer the evaluated object is to the evaluation criteria. Accordingly, the sensitivity value and adaptability value of the economic, social and ecological system of the 42 counties in the Wuling Mountains area can be calculated, and then the vulnerability value of the whole system can be attained.

### 2.4. Classification of Social-Ecosystem Vulnerability Levels

According to the existing research [49,50], the social-ecosystem vulnerability of the Wuling Mountains area is divided into five grades, namely very low, low, moderate, high and very high. The specific vulnerability index value domains are shown in Table 2.

### 2.5. Social-Ecosystem Vulnerability Impact Factor Identification

In order to identify the main obstacle factors and obstacle degree affecting the social-ecosystem vulnerability of the Wuling Mountains area, the index deviation degree (P_ij_), factor contribution degree (G_j_) and obstacle degree (I_j_) are introduced to calculate the impact of various indicators on the system vulnerability of the tourist regions. The obstacle degree model is calculated as follows:(8)$$\begin{array}{c} p_{\mathrm{ij}}=1-X_{\mathrm{ij}}^{\prime }\\ I_{j}=\left\{\left(G_{j}\times P_{\mathrm{ij}}\right)÷\overset{n}{\underset{i}{\sum }}\left(G_{j}\times P_{\mathrm{ij}}\right)\right\}\times 100\%\; \end{array}$$
where, X’_ij_ is the standardized value of the indicator, P_ij_ is the index deviation degree, namely the gap between the single index j and the optimal target value. G_j_ is the factor contribution rate, which is the influence of the single index j on the overall vulnerability of the system; it is generally expressed by the weight corresponding to the index. In addition, I_j_ is the obstacle degree that the item j index imposed to the vulnerability of the system.

## 3. Results

### 3.1. Timing Variation

#### Time Sequence Change in System Vulnerability

The overall comprehensive vulnerability of the system of the Wuling Mountains area presents a trend of “slow rise–steady decline”. It reached a peak of 0.7649 in 2014 and hit a minimum of 0.6273 in 2019. The vulnerability timing of each subsystem varies as follows:(1)The overall vulnerability of economic subsystem decreased slightly, but it was still in a highly vulnerable state. From Figure 2, the mean vulnerability index of economic subsystems in the Wuling Mountains area from 2010 to 2019 was 0.2766. Compared with social and ecological subsystem, the vulnerability of economic subsystem was still high. The overall vulnerability of the economic subsystem showed a trend of “steady decline and slow rise”. The vulnerability index decreased from 0.3003 to 0.2705, with an average annual decline of 1.12% and the lowest in 2014. The economic subsystem vulnerability index rose slowly from 2014 to 2019, ranging between 0.2635 and 0.2705, but was generally lower than the 2010 and 2012 index values.

(2)The vulnerability of the social subsystem has decreased slightly, with an average value of 0.2113, which was a moderate vulnerability. The overall inverted U-type, or “rise–steady decline”, trend evolved, with the maximum and minimum vulnerability index values of 0.2405 in 2012 and 0.1801 in 2019, respectively. It shows that the internal structure and function of the social subsystem in the tourist area is still sustainable. However, it is still in an unstable state. The system exposure-sensitivity degree is slightly high, and the adaptability affected by tourism needs to be further improved. From 2010 to 2012, the social subsystem vulnerability index increased continuously, with a high social subsystem vulnerability and an enhanced social subsystem instability. After 2012, the social subsystem vulnerability in this area was significantly reduced.(3)The vulnerability of ecological subsystem showed fluctuations, which increased from 2010 to 2014, and then decreased steadily. The mean value is 0.2265, and it is moderately vulnerable. It shows that the ability of tourist area to resist risk or tourism interference is unstable, the internal structure and function of the ecological subsystem are not robust, and the system structure still has room for optimization. The ecosystem vulnerability index dropped from 0.2433 in 2010 to 0.1857 in 2019, with an average annual decline of 2.63%.

### 3.2. Spatial Evolution

#### 3.2.1. Spatial Evolution of the Vulnerability of Each Subsystem

(1)Economic Subsystem

In Figure 3, the vulnerability of economic subsystem is generally high in the south and low in the north. The very highly and highly vulnerable areas are shifted from fragmented distribution to scattered distribution, and the vulnerability types are more diversified. In 2010, there were mainly very high, high and moderate types, but in 2019, there were all five types, mainly moderate vulnerability, meaning the vulnerability of economic subsystem had improved significantly. As shown in Table 3, from the perspective of provinces, the economic vulnerability varies from high to low respectively in Guizhou, Hunan, Hubei and Chongqing. In addition, the very highly and highly vulnerable counties are mainly distributed in the border areas of provinces (cities). Within each region, Wuchuan County in Guizhou province, Baojing County in Hunan province, Badong County in Hubei province, Xiushan County in Chongqing (before 2014) and Wulong District in the west (after 2014) are more vulnerable than other counties and cities in the region. The differentiation of the vulnerability of the county economic system in the four provinces (cities) is continuously expanding. The vulnerability of most counties and districts decreased, with Longshan County and Longhui County of Hunan Province, Xiushan County of Chongqing City, and Shiqian County of Guizhou Province decreasing significantly, while that in Wulong District increased. In addition, it has formed a stubborn and highly vulnerable cluster area with Guizhou’s Wuchuan County and Hunan’s Baojing County as the core.

(2)Social Subsystem

As shown in Figure 4, counties and districts with high vulnerability of social subsystem changed from centralized group distribution to scattered distribution in space, showing an overall downward trend. In 2010, the high vulnerability counties were mainly concentrated in Hunan and Guizhou, and later spread to four provinces (cities). In 2019, vulnerability focused on the border zone of provinces (cities). Low and very low vulnerability counties first decreased in number and then increased, shifting from northwest to the center and south. In 2019, in terms of vulnerability types, low and very low vulnerability counties accounted for 23.8%, moderate vulnerability counties 61.9% and highly vulnerable counties 14.3%, with no very high vulnerability counties. From the perspective of the distribution of provinces (cities), the social vulnerability varies from high to low level in Guizhou, Hunan, Hubei and Chongqing, respectively. Daozhen and Wuchuan counties in Guizhou, Huayuan and Fenghuang counties in Hunan, Zigui county in Hubei and Wulong counties in Chongqing are more vulnerable than other places in this area. Among them, the vulnerability of the social subsystem in Enshi City, Lichuan City and Zigui County in Hubei province, Sangzhi County and Fenghuang County in Hunan Province and Wulong District in Chongqing all increased, while the overall vulnerability of Guizhou area showed a downward trend, Xinhua County in Hunan province and Jiangkou County in Guizhou province decreased significantly.

(3)Ecological Subsystem

As shown in Figure 5, very high and high levels of the vulnerability of the ecological subsystem developed, overtime, from the west through the middle to the east, and the spatial structure of vulnerability evolved from group to block and ribbon. In 2019, extremely and highly fragile places were concentrated at the junction of Hunan, Hubei, Hunan and Chongqing. The number of very low and low degree vulnerability counties increased from dots to clumps, from two in 2010 to nine in 2019. From the perspective of the distribution of provincial (city) areas, Guizhou, Hunan, Chongqing and Hubei areas ranked from high to low vulnerability. Wuchuan, Dejiang, Yinjiang and other counties in Guizhou province and Baojing, Huayuan and Guzhang in Hunan province suffered a high vulnerability. The vulnerabilities of Shaoyang County in Hunan province, Pengshui County in Chongqing province and Dejiang County in Guizhou province had significantly reduced.

#### 3.2.2. Spatial Evolution of the Overall Vulnerability of the System

In Figure 6, the overall vulnerability of the system in the north of the study area is spatially better than in the south. From the distribution of provinces (cities), Guizhou, Hunan, Hubei and Chongqing suffered respective vulnerabilities from the highest to the lowest. Two stages, divided around the year 2014, were apparently self-evident. In the period of 2010–2014, the vulnerability generally increased, with the very high vulnerability mainly concentrated in the two blocks of Wuchuan County in Guizhou and Guzhang County in Hunan. The spatial structure evolved from the belt-like distribution of high vulnerability in the south and low in the north to the block-shaped high in the middle and low in the two wings. Wuchuan County, Guzhang County and Baojing County were constantly very highly vulnerable. In addition, their surrounding Daozhen County, Dejiang County, Luxi County and Chengbu County in the south had all developed from high vulnerability to very high vulnerability. Zigui County, Changyang County, Hefeng County, Fengdu County, Youyang County, Sangzhi County, Yongshun County, Anhua County and Tongdao County had changed from moderate vulnerability to high vulnerability, while other counties witnessed little change and fluctuated within the same vulnerability.

During the second stage, from 2015–2019, there were no very highly vulnerable counties, which had been transformed into high, moderate, low and very low vulnerability places, with the north part still superior to the south. Moderate and high vulnerability counties and districts were concentrated in the southern region. Low and very low vulnerability counties were seen in the northern part, forming two agglomeration areas of low and very low vulnerability in Hubei, Province with its Enshi City as the core, and in Chongqing Municipality with its Qianjiang District as the core. The nearby Lichuan City, Shizhu County and Pengshui County had all become low-vulnerable counties. In terms of the number of counties and districts, there were 22 moderate, low and very low vulnerability counties and districts in 2019, accounting for 52.38%. The number of very highly and highly vulnerable counties decreased from 32 in 2014 to 11 in 2019, with an average annual decline rate of 8.20%. Moderately and highly vulnerable counties gradually were transformed from very highly and highly vulnerable counties, and their vulnerability showed a general decline. Among them, the comprehensive vulnerability index of Qianjiang District touched the lowest level of 0.3827 in 2019.

### 3.3. Social-Ecosystem Vulnerability Impact Mechanisms

#### 3.3.1. Analysis of the Vulnerability Barrier Factors of Each Subsystem

The obstacle degree analysis model was introduced to calculate—according to Formula (8) in Section 2.4—the obstacle factors and obstacle degree of the vulnerability of the three subsystems in different years. The top three obstacle factors are regarded as the main factors, and the obstacle degree to the vulnerability of each indicator is evaluated, as shown in Table 4.

In the economic subsystem, the economic aggregate (D5), tourism revenue growth (D2) and proportion of total tourism revenue in GDP (D4) appeared most frequently and were selected as the main obstacle factors. The total tourism revenue (D1) was also frequent, and was chosen as a secondary obstacle factor of importance. The economic aggregate determines the economic strength of a county; in turn, the stronger the economic strength, the stronger the self-recovery ability when encountered with the external impact incurred via tourism. Tourism revenue growth, total tourism revenue and its proportion in GDP indicate that with the continuous development of tourism, the economic system is increasingly dependent on tourism, and the negative effect is increasingly prominent. The influence of tourism seasonality, emergency and instability hinders the declining vulnerability of economic subsystem in this area. It is of great significance to reduce the vulnerability of economic subsystems, stabilize the tourism revenue growth, promote the tourism transformation and upgrading of counties, and it is especially essential to attach importance to the tourism recovery in the post-epidemic era.

In the social subsystem, the urbanization rate (D11), tourist quantity (D9) and tourist population density (D12) are the main obstacle factors. Tourist number growth rate (D10) belongs to the secondary barrier factor. The four indicators are all at the exposure-sensitivity level, indicating that with the continuous development of tourism economy, the social conditions of the area have been constantly improved. The large number of foreign tourists and the construction of a wide variety of facilities greatly disturb the social subsystem. In order to reduce the vulnerability of social subsystem and realize the sustainability and stability of the system, it is particularly important to promoting the coordinated development of urban and rural areas, reduce the speed of urbanization and reasonably control the tourist number.

In the ecological subsystem, intensity of chemical fertilizer use (D18), pesticide use intensity (D19), area of artificial afforestation (D22) and environmental protection expenditure (D20) appeared the most frequently and stood the main obstacle factors, and land area covered with trees (D21) was a secondary obstacle factor. As the livelihood of farmers in this area belonged to the traditional agricultural planting type, it was highly dependent on land and the transformation of livelihood largely depended on land harvest. The use of high-intensity fertilizers and pesticides weakened the bearing capacity of rural environment in this area, and consequently increased the vulnerability of local ecological subsystem. Since 2014, the area of artificial afforestation and environmental protection expenditure had gradually replaced the intensity of chemical fertilizer use and pesticide use as the obstacle factors for the reduced vulnerability of ecological subsystem in this area. This indicated that the increase in artificial afforestation area and environmental protection expenditure had played a significant role in reducing the vulnerability brought about by the negative aspects of tourism. At the same time, the forest coverage rate also played an important part in reducing the vulnerability of the ecological subsystem. It is of great significance to conduct the following steps for improving the area’s ecosystem vulnerability: moving more farmers into tourism, increasing the diversity of farmers’ livelihood, improving the level of agricultural modernization including green agriculture, protecting the ecological environment and building an ecologically livable environment.

#### 3.3.2. Analysis of the Influence Mechanism of System Vulnerability

The social-ecosystem vulnerability of the Wuling Mountains area is the result of the interaction between system exposure-sensitivity and system adaptability. Among the system exposure-sensitivity elements, the factors affecting the reduction in system vulnerability mainly cover the tourism development and industrial production of economic subsystem, the tourism influence and social development of social subsystem, the demographic factors and agricultural development of ecological subsystem, which all have a positive relationship with relevant vulnerability levels. Among them, the obstacle degrees of total tourism revenue, tourism revenue growth, tourist number and urbanization rate remained at a high level from 2010 to 2019, indicating that with the development of regional tourism economy and urban construction, some counties in the Wuling Mountains still faced tourism interference. The obstacle degrees of rural population, intensity of chemical fertilizers and pesticides used against system vulnerability decreased from 49.37% in 2010 to 38.60% in 2019.

Among the adaptive capacity factors of the system, the factors affecting the reduction in system vulnerability mainly included the economic development and investment in economic subsystem, social expenditure on social subsystem, ecological background and investment in ecological subsystem, all of which had a negative relationship with the vulnerability level. Among them, the economic aggregate and total local fiscal revenue directly reflected the economic strength of the counties and districts. The initial time of tourism development and the strength of the development degrees rendered the economic situations different and disproportionate in some counties and districts, which is not conducive to the stability of regional systems in the long run. Compared with the education level and the number of medical beds, local financial expenditure and road density could more directly hinder the vulnerability decrease in the system, and the enhancement of the adaptability of social subsystem would directly affect the reduction in the overall vulnerability in the system. Since 2014, the area of artificial afforestation and environmental protection expenditure had gradually replaced the intensity of chemical fertilizer and the pesticide use. Moreover, the environmental protection expenditure had further hindered the reduction in the system vulnerability. It had become the obstacle factor for the reduced vulnerability of the ecological subsystem in this area. This was mainly because the counties in the Wuling Mountains area mainly used ecological expenditure for the treatment of wastewater and solid waste incurred from tourism development.

In general, the tourism social-ecosystem vulnerability in the Wuling Mountains area was closely related to tourism poverty alleviation and development. In terms of obstacle factors, the vulnerability reduction was mainly contained by the rising exposure-sensitivity of economic and social subsystems brought about by the development of tourism economy. Meanwhile, due to the increase in economic development and social expenditure generated by tourism development, the adaptability of the system had increased, and the improved and decent ecological conditions in the region were conducive to the system stability. Compared with 2010, the vulnerability of social ecosystem in the Wuling Mountains area were effectively reduced, leaving the internal structural stability of the system still on the way to further improvement and optimization.

## 4. Discussion

### 4.1. Analysis of the Temporal Variability of the Vulnerability

From 2010 to 2019, the overall comprehensive system vulnerability of the Wuling Mountains area showed a trend of “slow rise–stable decline”. In 2010, most counties and districts began to pay attention to the development of tourism, and the number of tourists gradually increased. However, the ruthless exploitation brought about by tourism development had led to the increasing environmental pollution and ecological crisis [29]. Additionally, the high-level innate vulnerability of the system rendered the overall vulnerability of the area at a high degree. With the implementation of the “targeted poverty alleviation” strategy in 2013, under the background of strong support from the national poverty alleviation policy, the poverty alleviation and development of the tourism industry was improving rapidly. The development mode of social and economic structure was changed, the number of tourists was on the rise, and the overall system development was under full swing [51].

Overall, during the stipulated period, the vulnerability of economic subsystem decreased slightly, but remained high. The economic driving effect of tourism posed on the research area was relatively obvious [52], and the economic benefit generated by tourism toward poverty alleviation was significant. However, the internal structure and function of the economic system fluctuated violently with a low stability, and the capability of self-recovery from the influences by tourism remained weak. With the improvement of traffic conditions and the exploitation of tourism resources, the proportion of total tourism revenue in GDP and the dependence of the economy on tourism were constantly increasing, which, in turn, led to the slow increase in the vulnerability index of the economic subsystem [53]. Specifically, the economic vulnerability of Longshan County and Longhui County in Hunan province, Xiushan County in Chongqing province and Shiqian County in Guizhou province had decreased significantly. Longshan County and Longhui County led the overall economic and social development with the help of targeted poverty alleviation there. By relying on the beautiful natural scenery and endowed ecological resources, they had vigorously implemented the industrial poverty alleviation, thus accelerating the integrated development of cultural, ecological and tourism-characteristic industries. They strived to increase the income of local residents through the development of leisure and sightseeing agriculture, as well as the manufacturing of characteristic products and the tourism service industry. Chongqing’s Xiushan county adhered to the fundamental policy of poverty alleviation for industries by means of constructing the “characteristic agricultural demonstration base in the Wuling Mountains”, it had endeavored toward the goals of “each village equipped with industrial programs, and every household mastering certain employment skills.” To be more specific, this county mainly focused on the development of such five industries as traditional Chinese medicine (TCM) tea, walnut, quality fruits and ecological husbandry, thus constantly consolidating the county’s alleviation achievements. Based on its resource advantages, Shiqian County of Guizhou Province had vigorously developed the “ten industries for poverty alleviation “, especially the three leading alleviation industries of tea, medicine and fruit. For that effect, it is advised that, in the subsequent rural revitalization strategy, the diversification of industries shall be encouraged in order to reduce the vulnerability of the economic subsystem.

From 2010 to 2012, the social vulnerability index was constantly rising. During this period, the tourism industry in the Wuling Mountains area began a steady development. Nevertheless, due to its small number of tourists, the large gap between regional urbanization rates, especially the low rural employment rate and the overall low consumption level, the high vulnerability of social subsystem and the rising system instability occurred [54]. After 2012, with the implementation of the poverty alleviation policy, a large number of policies and social funds were introduced into the area, and a larger proportion of local fiscal revenue were budgeted for social infrastructure construction. Moreover, investments in medical care, education and transportation were reinforced [55]. As a result, educational spending increased from 12.12862 billion yuan in 2012 to 34.780.53 billion yuan in 2019, the number of medical beds expanded from 54,751 to 121,502, with traffic access rate rising year by year. The adaptability of the social subsystem in this area was also enhanced, and the positive effect of tourism poverty alleviation on social system was fully generated. For instance, the social vulnerability of Xinhua County in Hunan province and Jiangkou County in Guizhou province had decreased significantly. Xinhua is the reservoir-region county with the most migrants in Hunan province. The local government highly valued the subsequent task of helping reservoir migrants, focusing on constructing beautiful villages in the reservoir region. They also tried to improve the livable environment for migrants, observing rural cultural customs and basing social harmony and local stability on the construction of traditional culture and rural civilization. Guizhou’s Jiangkou county is the birthplace of the national targeted poverty alleviation registration system, and also the birthplace of the national poverty prevention mechanism of monitoring, precaution and guarantee. Since 2012, Jiangkou County had built a comprehensive transportation system, with national and provincial trunk roads and county and township road networks in all directions. The government had actively invested in education and made great effort to ensure the schooling for children of urban and rural residents. Various water conservancy projects and drinking-water improvement programs had also been implemented; telecommunications network rendered stable and smooth service with 4G networks fully accessible. They had also implemented the “four-key medical care” policy to ensure basic medical assistance for all, and provided one-stop settlement windows to facilitate payment and reimbursement for everyone in need of medical service. Rural housing projects had also been carried out, with more than 48,000 rural households in the county having had their endangered houses renovated. It is safe to conclude, therefore, that significantly effective means to reduce the vulnerability of the social subsystem are for the government to vigorously invest in infrastructure construction to ensure the needs of urban and rural residents for housing, medical care and education, as well as for the improvement of their living environment.

The vulnerability of ecological subsystem fluctuated from 2010 to 2014, and then steadily declined. Since the issue of *The Regional Development and Poverty Alleviation Plan of The Wuling Mountains Area (2011–2020)*; this area has adhered to the purpose of ecological protection and green development. It focuses on implementing ecological greening projects such as ecological migration and returning farmland to forests. The local governments have attached great importance to ecological issues, and taken a series of positive measures to improve the environment. Their concrete steps include increasing energy conservation and environmental protection expenditure, reducing the use of pesticides and fertilizers, adjusting the industrial structure and vigorously controlling sewage discharge [56]. The vulnerability of ecological subsystems in Shaoyang County of Hunan province, Pengshui County of Chongqing Province and Dejiang County of Guizhou Province has been greatly reduced. Among them, Shaoyang County has practiced the concept of ecological priority and green development, promoted the construction of ecological civilization and environmental protection, and continuously improved the ecological and environmental quality. Shaoyang has so far formulated a number of local environmental laws and regulations to promote the improvement of urban ecology and living environment. At the same time, it has continuously increased its investment in infrastructure construction and environmental protection, with its pollution prevention and control capacity greatly improved. Dejiang County in Guizhou Province has based its endeavors on the guidance of elaborate planning, and has compiled and implemented a series of ecological protection plans, such as the *13th Five-Year Plan for Environmental Protection*
*in Dejiang County* and the *Comprehensive Environmental Pollution Control Plan*
*in Dejiang County*. It has increased the financial input in various ecologically innovative activities, and formulated incentive policies for ecological construction, with a priority on the creation of awards, demonstration project subsidies, publicity and education. Moreover, it has mobilized all authorities to actively participate in the construction of ecological civilization, and advocated the public to engage in the construction of the “Beautiful Dejiang”. Therefore, the formulation of reasonable environmental protection laws and regulations, investment in environmental protection funds, and involvement of the general public in all the stated undertakings are desirable measures to reduce the vulnerability of ecological subsystems.

### 4.2. Analysis of the Spatial Variability of the Vulnerability

Excessive reliance on tourism often adversely affected on regional economic development [57]. In the early stage of the tourism development in Wulong District, the momentum was fast and strong, but the tourism quality had lingered at a low level, indicating that the tourism industry structure needed modification and improvement. The dependence of local economy on tourism benefits increased, the economic risks increased and so did the vulnerability. Baojing County, which is located in the mountains, had a rather weak economic foundation. In 2014, the county focused on the development of tourism economy, but the utilization of its resources stood at a low level, the driving effect on the local economy was not obvious, and its economic vulnerability remained high. At the same time, as China’s regional economy was still running according to the law of ”administrative economy”, the administrative boundary stood like an “invisible wall” and imposed rigid constraints on the horizontal connection of regional economy [58]. The provincial border regions operated separately and independently, resulting in slow economic development, weak coordination for and sustainability of regional development, and a vulnerable economic system.

Weak infrastructure has always been a bottleneck restricting the development of Guizhou Province. Poverty alleviation and development through tourism have made remarkable achievements in railway, highway, civil aviation and water transportation construction in the Wuling Mountains’ Guizhou area. Hubei’s Enshi City, Lichuan City and other counties and districts have a good economic foundation, improved transportation accessibility and relatively perfect medical and education facilities. However, the influx of tourists has led to the gradual increase in immigrant population and tourist density, thus the conflict between tourists and local residents has intensified [59], and so has the pressure of social subsystems. Generally, the overall vulnerability of the social subsystems is on the decline, but it still needs to be further improved.

Ecological vulnerability varies from high to low in Guizhou, Hunan, Chongqing and Hubei, respectively. Wuchuan, Dejiang, Yinjiang and other counties in Guizhou province and Baojing, Huayuan and Guzhang in Hunan province have a high vulnerability. Nine counties, including Qianjiang District, Shaoyang County, Enshi City, Lichuan City, Pengshui County, Sinan County, Jiangkou County, Xinhua County, Xinhua County and Anhua County, suffer a low ecological vulnerability. Environment-friendly tourism development modes such as ecotourism and rural tourism have been providing a guarantee for the ecological environment improvement in these counties. Their ecological levels are higher than those of other counties, and the ecosystems in these counties are highly adaptable to resist the risks brought about by tourism development. However, Wuchuan and Guzhang counties have long been the core of two agglomerate regions with high vulnerability. These deeply impoverished counties have been successfully lifted out of poverty, but only with the help of strong policy support. Most farmers there are still in favor of traditional farming; the agricultural technology maintains a weak foundation; and the already barren soil is suffering intensive use of fertilizers and pesticides. All these may well corner the ecological environment system in an awkward high pressure. Consequently, the system bears a weak capability of self-recovery, with the ecosystem vulnerability lingering at a high level.

### 4.3. Analysis of Comparisons with Similar Studies

The results showed that the vulnerability of social ecosystem in the Wuling Mountains area showed a trend of “slow rise—stable decline”, and it is still in the moderate vulnerability level, which is similar to the research findings in other less developed areas. Chen, J. found that the vulnerability of the 14 administrative villages at the northern foot of the Qinling Mountains was generally moderate after tourism development [60]. Cui, X. believed that the social ecosystem of Qinling-Bashan Mountainous areas was in a moderate vulnerable state, and the ability of the system to resist risks and shocks was still insufficient [61]. The vulnerability index of the economic subsystems was the highest in the internal structure of the system vulnerability in the Wuling Mountains area, which was different from the conclusion that the ecological subsystems was the most vulnerable in the study of similar areas by Jia, Y. and Wang, Q. [62,63]. In terms of the influencing factors of overall regional vulnerability, Wang, Q.’s conclusions based on the poor mountainous area-Dabie Mountains area as the research object were basically consistent with this study [64]. At the same time, there are similarities in the main obstacles to the reduction in vulnerability of subsystems in similar studies [51]. This suggested that there are commonalities between the main factors affecting the reduction in social-ecosystem vulnerability in poor mountain destinations. Comparing and studying the vulnerability of different tourist destinations in similar poverty-stricken mountainous areas, and comprehensively sorting out the influencing factors of the vulnerability of social ecosystems in such areas, the practical path of reducing vulnerability can be further summarized, and the practical guidance and universal applicability of theoretical references can be increased.

### 4.4. Policy Implications

This paper integrated the theory of vulnerability and the analysis framework of social ecosystem into tourism research, and analyzed the evolution law of social-ecosystem vulnerability in time and space from the county-scale evaluation. At the same time, the influence of tourism development on the poverty-stricken mountainous regions were discussed, which is expected to help promote the coordinated development of tourism and ecological protection in the Wuling Mountains area. Moreover, the cases provided are likely to support for the goals of realizing rural transformation, regional sustainable development and consolidating the effective connection between the achievements of poverty alleviation and rural revitalization. It is of great practical significance to solve the contradiction between tourism development and natural ecology in poverty-stricken mountainous regions, and to realize the coordinated development of human and land system in the Wuling Mountains area.

Poverty alleviation and rural revitalization are a systematic and long-term process. With the help of tourism poverty alleviation and ecological relocation, 42 key counties and districts in the Wuling Mountains area have been successfully lifted out of poverty. However, they are still in the primary stage of rural revitalization, and there is an obvious imbalance in the development in the area. Very high vulnerability counties are mainly concentrated in Guizhou’s Zunyi and Hunan’s Xiangxi, while moderate and low vulnerability counties are concentrated in Hubei and Chongqing. The agricultural foundation of Hubei and Chongqing is relatively stable and belongs to the strategic scope of the Yangtze River Economic Belt. The economic scale, industrial development, education, medical care and people’s livelihood services in the two counties are relatively sound, and the they have strong adaptability to resist external risks. Guizhou and Hunan are typical karst landform regions, with rugged surface, large altitude difference, barren soil, weak agricultural foundation, poor transportation accessibility, high vulnerability of the native system, deep poverty level and weaker economic foundation. The vulnerability of the social subsystem in Chongqing and the ecological subsystem in Guizhou has shown an upward trend, and the economic subsystem of the Wuling Mountains area is still in a highly fragile state. Despite the fact that the tourism in Chongqing started at an early age, and that the number of tourists and the amount of tourism revenue in recent years have increased rapidly, the large influx of external tourists may well bring about social resource tensions and the impact of alien cultures on the local ethnic minority cultures [65]. Meanwhile, attention should be paid to the negative social impact caused by the uneven distribution of income in the tourism industry [22]. Therefore, local governments should provide different policy support during the rural revitalization stage to promote the sustainable and stable development of the Wuling Mountains area. Additionally, each region concerned needs to take targeted improvement measures based on its own facts.

It should be emphasized that sustainable development in the longer period still depends on both short-term measures and medium- and long-term strategies. In the short run, necessary measures should be taken to reduce social-ecosystem vulnerability according to the main vulnerability factors of subsystems in different regions. (1) As for the economic subsystem, such indices as the economic aggregate (D_5_), the tourism revenue growth rate (D_2_) and the proportion of total tourism revenue in GDP (D_4_) are the main obstacle factors. Counties and districts with high vulnerability should do their utmost to improve the business environment, and attract investment from quality enterprises. They should also increase capital investment, improve infrastructure and enhance the introduction effects of capital, technology and talents. They are also encouraged to grant preferential tax support to key industries, actively attract foreign investment, promote economic growth, facilitate scientific and technological progress and boost the development of high-tech industries. (2) As for the social subsystem, urbanization rate (D_11_), tourist quantity (D_9_) and tourist population density (D_12_) are the major obstacle factors. Counties and districts with high vulnerability should continue to increase investment in infrastructure construction, expand the coverage of education, medical care, vocational training and social security, a combined effort to upgrade the level of urbanization. They should further optimize service facilities for the tourism reception, strengthen the reception capacities of transportation, accommodation, catering, entertainment and other services, as well as the tourist reception capacity, and meet the needs of tourists and local residents. (3) As for the ecological subsystem, intensity of chemical fertilizer use (D_18_), pesticide use intensity (D_19_), area of artificial afforestation (D_22_) and environmental protection expenditure (D_20_) are the chief barrier factors. Counties and districts with high vulnerability should stress the publicity of environmental protection, enhance farmers’ environmental awareness and encourage them to reduce the use of pesticides and fertilizers. At the same time, the ecological compensation work should be well conducted to reduce the economic losses from farmers. They should increase investment in environmental protection, and implement environmental protection projects such as afforestation and returning farmland to forests.

Attention shall also be paid to the medium- and long-term perspective of strategies. (1) As for the border zones of the four provinces (cities), we should not be confined by the restrictions of administrative boundaries. Instead, we should surpass the administrative boundaries [66], strengthen the communication and cooperation between provinces and regions in the economic field, establish a good organization and coordination mechanism. We should also reasonably distribute development interests, integrate superior resources, and realize the mutually beneficial sharing of talents, technology and resources [67]. In the overall tourism development of the Wuling Mountains area, we should give full play to the economic driving role of Hubei and Chongqing by stressing the coordination and collaboration of the four neighboring provinces for a coordinated development of regional tourism industry. (2) We should encourage and educate the residents in the mountainous areas to be familiarized with alien cultures, and improve their own spiritual and cultural literacy. We also should maintain the characteristic advantages of traditional ethnic culture, promote the integration of local and incoming cultures [68] and shape civilized and progressive local customs and practices. We should improve the income distribution mechanism for the tourism industry to help residents in poverty-stricken zones achieve common prosperity. We should also improve the social governance system and governance capacity toward the goal of effective local governance. (3) As the ecological environment of Guizhou is very vulnerable, a long-term ecological environment protection mechanism should be established to create a decent ecological and livable environment. We should accelerate the construction of a standardized compensation system for tourism ecological protection, and establish a coordination ecological compensation mechanism for inter-provincial tourism. Furthermore, we can encourage the integrated development of agriculture, forestry, culture, marine and other related industries or businesses [69]. Additionally, we should increase the weight and proportion of environmental protection factors in approving tourism project applications, introduce environmentally friendly enterprises and focus on the layout of new tourism industries such as those specialized in ecological restoration and landscaping [70]. As the COVID-19 pandemic has instilled more rational and healthier consumption concepts in tourists [71], health tourism, ecological tourism and field-research tourism are expected to become the future mainstream. It is thus essential to build green, nursing and ecological tourist destinations while protecting and utilizing ecological and environmental resources such as the beautiful mountains and rivers in Guizhou. (4) Some typically vulnerable counties, such as Youyang County in Chongqing and Wuchuan County in Guizhou, were listed in 2021 as key supported counties for China’s rural revitalization. These counties and districts should seize the opportunity to strengthen their weaknesses in development, and comprehensively consolidate and expand the achievements of poverty alleviation. Chongqing’s Youyang County, for its high economic vulnerability, should further optimize the industrial structure and improve the “tourism +” industrial system. Moreover, the vulnerability of the three subsystems in Guizhou’s Wuchuan County stands at a high level. Policy support for the county should be comprehensively strengthened, resource sharing among counties should be tilted toward it, and the County itself should make active efforts to speed up its development. (5) In the stage of rural revitalization, we should actively promote the coordinated development of tourism industry and other related industries, and develop diversified industrial structure to promote high-quality development of regional economy. This could facilitate industrial prosperity, prevent local economy heavily relying on tourism, and avoid the potential “Dutch disease” [72]. At the same time, we should give full play to the tourism multiplier effect, drive economic growth through the tourism industry, create more jobs and increase residents’ income [52]. Moreover, we should play a comprehensive role in promoting the effective connection between poverty alleviation and rural revitalization, so as to better boost rural revitalization. (6) Due to the influences of the COVID-19 pandemic, local governments should also take actions to adopt measures conducive to enhancing human health and advancing the sustainable development goals. Efforts should also be made to accelerate the necessary transformation of economic recovery, and to develop low-carbon, affordable energy and sustainable transportation systems for long-term economic and social benefits. (7) Tourism enterprises and other various enterprises in the Wuling Mountains area should actively assume their social responsibilities, support art and educational institutions, plus health and medical undertakings and participate in environmental protection and biodiversity promotion. This will not only help further reduce social-ecosystem vulnerability, but it will also be important in improving profits, productivity and performance [73]. For each industry, there are specific sustainable development goals (SDGs), and on each SDG there are specific industries which exert high impacts. Some sectors are generally more important to SDGs than others [74]. The setting of business goals should adapt to the regional sustainable development strategy. NGOs such as business associations should establish appropriate assessment mechanisms and standards to better measure the contribution of enterprises in such domains as sustainable development, ecology, society and governance and guide various enterprises to better help realize social well-being—a move toward the direction of sustainable development [75].

This paper still needs to improve the following aspects. (1) When elaborating the research objects, this paper only selected and analyzed 42 counties in the Wuling Mountains area, which are likely to cast some limitations on the research results and weaken the general reference significance of the research conclusion. In the future, all the 71 counties (districts) of the four provinces (cities) in the Wuling Mountains area could be selected for comparative analysis. (2) This paper only identified the influencing factors of social-ecosystem vulnerability in the Wuling Mountains at the macro scale, which might reduce the accuracy of the vulnerability evolution mechanism. Therefore, in the future, horizontal comparative analysis shall be carried out in terms of the four provinces (cities), individual counties, communities, or farmers, as well as other different scales, so as to explore the action mechanism of the influencing factors more comprehensively. Consequently, we can apply targeted policies to different types of tourist areas, and better guide the sustainable development of tourism according to local conditions. (3) This study selected data only before the COVID-19 pandemic outbreak in 2020, but the COVID-19 global crisis has greatly impacted the evolution of social-ecosystem vulnerabilities in the area of the Wuling Mountains, and the achievement of the sustainable development goals in the regional cluster. The deadly pandemic is case in point that human and nature are closely related and interactive. In addition, strong evidence suggests that people who enjoy clean air and drinking water are more likely to withstand the outbreak than those who suffer from air pollution or lack clean drinking water. In follow-up studies, we will focus on the impact of the COVID-19 pandemic on local sustainability and its assessments, providing a scientific theoretical reference for the development of the region in the “post-pandemic era”.

## 5. Conclusions

To sum up, our work has established a close connection between the law of the spatial-temporal change in the social-ecological system in the Wuling Mountains area and its main influencing factors, which provides a scientific policy reference for the high-quality development of tourism, rural revitalization and common prosperity in region of the Wuling Mountains. With the help of SES theory and vulnerability analysis framework, this study constructed an index evaluation system from the aspects of exposure-sensitivity and acclimatization. By means of the data collected from field interviews and questionnaire surveys, the vulnerability level of the social ecosystem in the Wuling Mountains area and its spatial-temporal evolution law in 2010–2019 were measured, and the obstacle factors and influence mechanisms of vulnerability were analyzed and detected.

As for the initial goals proposed at the beginning of this study, it can now be noted that we have successfully clarified the vulnerability change characteristics of the economic, social and ecological subsystems in both temporal and spatial dimensions of the study area, providing insights into the barrier factors and impact mechanisms of reduced social-ecosystem vulnerability. Firstly, the scope of the current study is to determine the variation of social-ecosystem vulnerability in the Wuling Mountains. Secondly, this study aimed to identify the major barrier factors for reducing social-ecosystem vulnerability in the chosen mountainous area. Finally, one of the main objectives of this scientific paper is to highlight what measures the local governments should take to reduce the social-ecosystem vulnerability of the Wuling Mountains. Moreover, it should be emphasized that we believe that our results are a reference for all backward mountainous areas in China. This study has shown that, due to the gradual development of poverty alleviation through tourism, the vulnerability of the social ecosystem in the Wuling Mountains area has undergone complex and profound changes under the action of various internal and external factors. Moreover, a positive conclusion can be drawn from our study that poverty alleviation through tourism has indeed reduced the local social-ecosystem vulnerability to some extent. From the years 2011–2019, the overall vulnerability showed a trend of “slow rise–steady decline”, and the vulnerability of each subsystem decreased to varying degrees. In terms of timing changes, the total system vulnerability peaked in 2014 and had since decreased year by year. In 2012, China made it clear that it would have achieved the grand goal of completing the building of a moderately prosperous society in all respects by 2020. What is more, it clarified such basic tasks for the 2020 ambition as lifting the rural poor out of poverty as scheduled, and addressing regional poverty. In 2013, the state began to implement the “targeted poverty alleviation” strategy, and the government’s poverty alleviation endeavor has played a significant role in reducing the social-ecosystem vulnerability in the region. The social ecosystem showed a trend of “slow rise–steady decline”, but it was still moderately vulnerable. Among them, the overall economic subsystem decreased slightly and was still highly vulnerable, the social subsystem declined slightly and the ecological ecosystem fluctuated and declined steadily after 2014.

The study on the spatial change in social-ecosystem vulnerability in the Wuling Mountains area shows that the overall vulnerability of the system is better in the north than that in the south, and the vulnerability of counties in the border zone of provinces (cities) is generally high. From the perspective of spatial evolution, the vulnerability of the four provinces (cities) runs from high to low: Guizhou, Hunan, Hubei and Chongqing, with the provinces in the southern area bearing a significantly higher vulnerability. Among the 42 counties, the areas with a higher vulnerability are mainly distributed in the provincial (municipal) border zones, which is more prominent in terms of the economic subsystem.

The present findings also point to the barrier factors hindering social-ecosystem vulnerability in the area. In general, the social-ecosystem vulnerability of tourist areas in the Wuling Mountains area is the result of the interaction of system exposure-sensitivity and system adaptability. Economic aggregate, tourism revenue growth rate and the proportion of tourism revenue in GDP have the biggest impact on the vulnerability of economic subsystem. The stronger the economic strength, the stronger the system’s self-resilience to external shocks caused by tourism. The continuous development of tourism will lead to the dependence of the economic system on tourism. However, tourism is greatly affected by seasonality, emergency and instability, which hinder the decline of economic subsystem vulnerability in this area. The urbanization rate, the number of tourists and the tourist population density have the greatest impact on the vulnerability of the social subsystem. Specifically, with the rapid development of tourism and the influx of tourists into the region, it is bound to cause tourist-host encounters. Tourists will share local social resources such as transportation, accommodation and medication. Once the number of tourists exceeds the local load-bearing capacity, the local facilities will fail to meet the needs of both the tourists and local residents, which is likely to cause social unrest such as dissatisfaction and conflict among local residents. As a possible solution, strengthening infrastructure development can improve the local tourist reception capacity, increase its tolerance for extra tourists, and ease the conflicts or contradictions between the hosts and guests. The intensity of chemical fertilizer use, pesticide use, artificial afforestation area and environmental protection expenditure can exert the greatest impact on the vulnerability of ecological subsystems. Since most of the residents in the Wuling Mountains area still live on traditional agriculture, pesticides and fertilizers are the main sources of pollutants. Increasing the area of artificial afforestation and expenditure on environmental protection might be the most direct and effective means for the protection of the local environment.

## Figures and Tables

**Figure 1 ijerph-19-11688-f001:**
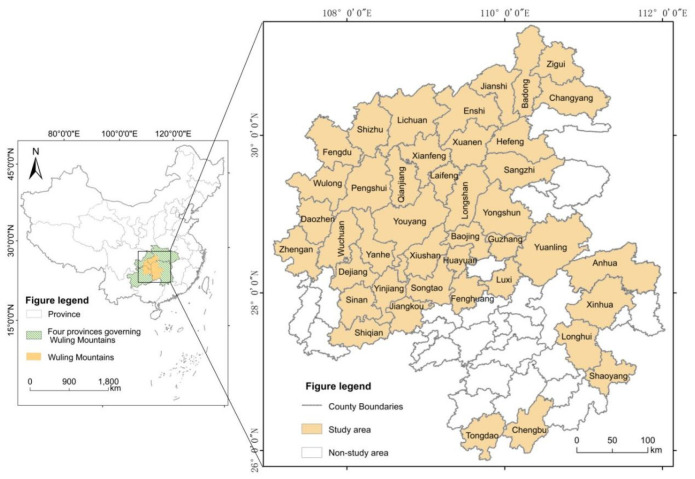
The scope of the study area.

**Figure 2 ijerph-19-11688-f002:**
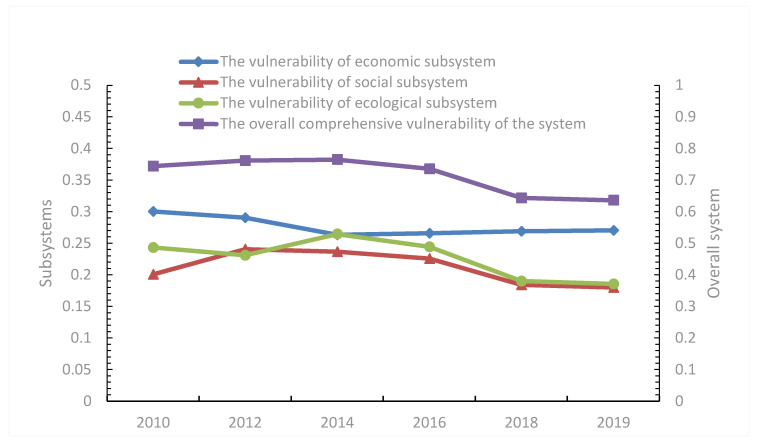
Time sequence change diagram of the vulnerability of the system and three social-ecological subsystems of the tourist areas in the Wuling Mountains.

**Figure 3 ijerph-19-11688-f003:**
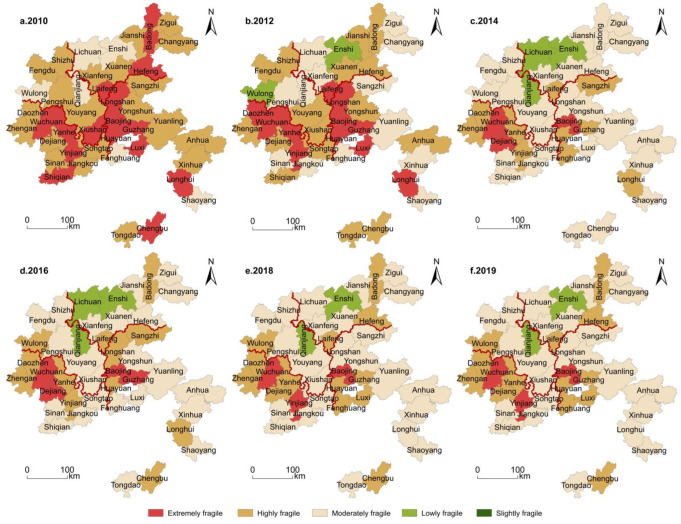
Spatial differentiation of economic subsystem vulnerability of tourist areas in the Wuling Mountains.

**Figure 4 ijerph-19-11688-f004:**
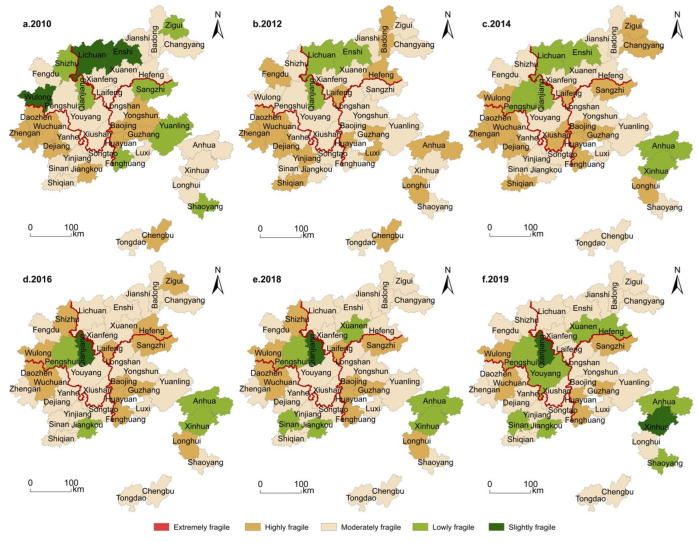
Spatial differentiation of social subsystem vulnerability of tourist areas in the Wuling Mountains.

**Figure 5 ijerph-19-11688-f005:**
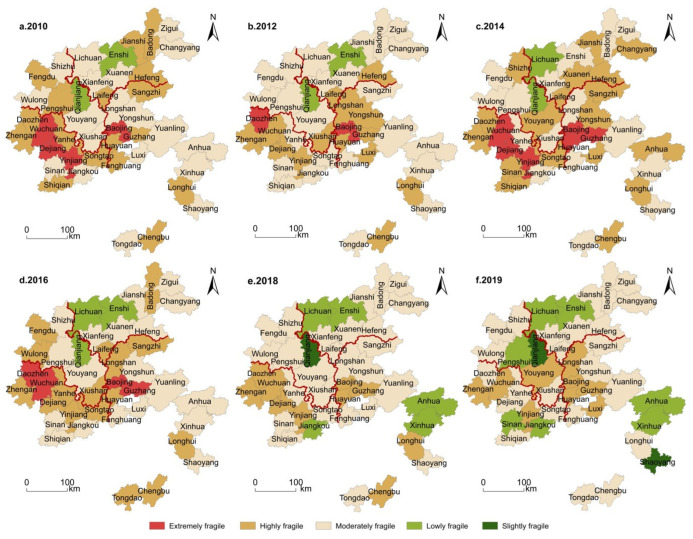
Spatial differentiation of the ecological subsystem vulnerability of tourist areas in the Wuling Mountains.

**Figure 6 ijerph-19-11688-f006:**
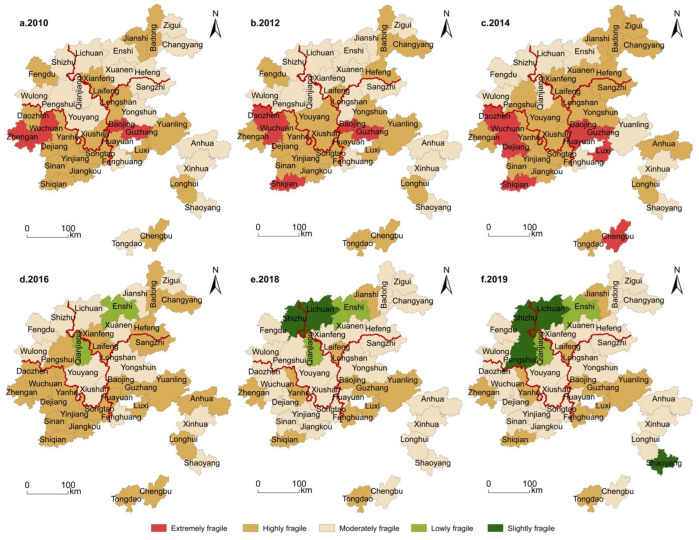
Spatial differentiation of the social ecosystems in the Wuling Mountains.

**Table 1 ijerph-19-11688-t001:** Vulnerability evaluation index system for tourism-based social-ecological systems under SEE and VSD.

System Layer	Criteria Layer	Index Layer	Unit	Code	Indicator Description and Property	References	Weight
Economic subsystem vulnerability	Exposure- sensitivity	Total tourism revenue	10^4^ yuan	D_1_	Reflects economic benefits of local tourism (+)	[43]	0.0674
		Tourism revenue growth rate	%	D_2_	Reflects the pressure brought about by the tourism economic growth (+)	[44]	0.0560
		Tourism economic density	10^4^ yuan/km^2^	D_3_	Reflects the economic pressure, namely the tourism income/regional area (+)	[45]	0.0554
		Proportion of total tourism revenue in GDP	%	D_4_	Reflects the dependence of the local economy on tourism (+)	[46]	0.0769
	Acclimatization	Economic aggregate	10^4^ yuan	D_5_	Reflects the overall economic response capacity, (−)	[41]	0.0390
		Total local fiscal revenue	10^4^ yuan	D_6_	Reflects the economic strength of the government, (−)	[41]	0.0329
		Growth rate of fixed asset investment	%	D_7_	Reflects the capital investment intensity (−)	[41]	0.0318
		Per capita GDP	10^4^ yuan	D_8_	Reflects the level of resident disposable income (−)	[41]	0.0230
Social subsystem vulnerability	Exposure- sensitivity	Tourist quantity	10^4^ people	D_9_	Measures the pressure from tourists (+)	[47]	0.0641
		Tourist number growth rate	%	D_10_	Measures the pressure from tourist growth (+)	[41]	0.0606
		Urbanization rate	%	D_11_	Measures the local urbanization level (+)	[41]	0.0325
		Tourist population density	people/km^2^	D_12_	Measures the population structure, namely the number of tourists/the number of local residents, (+)	[41]	0.0623
	Acclimatization	Local financial expenditure	10^4^ yuan	D_13_	Measures the local social financial capacity (−)	[41]	0.0346
		Number of medical beds	\	D_14_	Measures medical security capacity, that is, with the number of hospital beds (−)	[14]	0.0306
		Educational level	10^4^ yuan	D_15_	Reflects the local education support strength, namely the education expenditure (−)	[14]	0.0386
		Road density	km/km^2^	D_16_	Reflects accessibility of the local road (−)	[48]	0.0342
Ecological subsystem vulnerability	Exposure- sensitivity	Total rural population	10^4^ people	D_17_	Measures the pressure of the rural ecological environment (+)	[47]	0.0365
		Intensity of chemical fertilizer use	kg/hm^2^	D_18_	Measures the pressure of chemical fertilizer on the ecological environment (+)	[47]	0.0567
		Pesticide use intensity	kg/hm^2^	D_19_	Measures of pesticide pressure on the ecological environment (+)	[47]	0.0714
	Acclimatization	Environmental protection expenditure	10^4^ yuan	D_20_	Reflects the local ecological environment improvement efforts (−)	[41]	0.0352
		Land area covered with trees	km^2^	D_21_	Reflecting the natural conditions of the local (−)	[41]	0.0301
		Area of artificial afforestation	km^2^	D_22_	Reflects the local ecological environment protection strength (−)	[41]	0.0302

Note: + means the indicator is positively related to system vulnerability, and − shows a negative correlation.

**Table 2 ijerph-19-11688-t002:** Classification and characteristics of social-ecosystem vulnerability in the Wuling Mountains.

Vulnerability Classification	Vulnerability System	Vulnerability Index Values	Description
Very low	Economic	(0.166–0.193]	The social ecosystem of tourist area is in a relatively stable and ideal state. The system has a strong adaptive management capability and can operate effectively to maintain the balance of the system.
	Social	(0.115–0.157]	
	Ecological	(0.133–0.171]	
	Comprehensive	(0–0.40]	
Low	Economic	(0.193–0.230]	The social ecosystem of tourist area has strong anti-interference capability and high adaptability, which promotes the system to effectively and quickly return to the balance position of the system when it encounters risks.
	Social	(0.157–0.199]	
	Ecological	(0.171–0.209]	
	Comprehensive	(0.40–0.55]	
Moderate	Economic	(0.230–0.267]	When the social ecosystem of the tourist area is faced with external risks and interference, it has moderate repair and adaptation capability, and have a relatively stable external impact, which reduces the exposure-sensitivity to a certain extent. However, if the risk or interference increases, the system will still evolve towards extreme vulnerability.
	Social	(0.199–0.241]	
	Ecological	(0.209–0.246]	
	Comprehensive	(0.55–0.70]	
High	Economic	(0.267–0.305]	The social ecosystem of tourism area has high exposure-sensitivity, and the system recovery and adaptability are weak. Potential risks will have a huge pressure and impact on the system; the system instability is strong.
	Social	(0.241–0.283]	
	Ecological	(0.246–0.284]	
	Comprehensive	(0.70–0.85]	
Very high	Economic	(0.305–0.332]	The social ecosystem of the tourism area has high exposure-sensitivity characteristics. When the system is faced with external disturbance and pressure, its recovery and adaptability are very weak. If unstable factors continue to intensify, the system will stagnate or even collapse in a short time.
	Social	(0.283–0.324]	
	Ecological	(0.284–0.323]	
	Comprehensive	(0.85–1.00]	

**Table 3 ijerph-19-11688-t003:** Social-ecosystem vulnerability of tourist areas in the four provinces (cities) of the Wuling Mountains.

Region	Subsystem	Mean	Region	Subsystem	Mean
	Economic subsystem	0.2618		Economic subsystem	0.2805
Hubei area	Social subsystem	0.1997	Hunan area	Social subsystem	0.2319
	Ecological subsystem	0.2134		Ecological subsystem	0.2349
	Economic subsystem	0.2558		Economic subsystem	0.2932
Chongqing area	Social subsystem	0.1930	Guizhou area	Social subsystem	0.2501
	Ecological subsystem	0.2181		Ecological subsystem	0.2421

**Table 4 ijerph-19-11688-t004:** Social-ecosystem vulnerability barrier factors in the Wuling Mountains Area.

Year	Class	Economic Subsystem			Social Subsystem			Ecological Subsystem		
2010	Obstacle factors	D_5_	D_6_	D_8_	D_11_	D_9_	D_15_	D_17_	D_18_	D_19_
	Obstacle degrees	17.24%	16.43%	15.15%	14.90%	13.53%	12.36%	21.67%	19.39%	18.05%
2012	Obstacle factors	D_1_	D_5_	D_2_	D_13_	D_11_	D_10_	D_19_	D_18_	D_21_
	Obstacle degrees	18.66%	16.13%	14.01%	15.62%	12.39%	11.21%	22.10%	18.92%	17.41%
2014	Obstacle factors	D_2_	D_3_	D_4_	D_9_	D_11_	D_12_	D_20_	D_19_	D_22_
	Obstacle degrees	17.71%	16.32%	15.09%	12.80%	10.95%	10.17%	22.04%	19.18%	18.37%
2016	Obstacle factors	D_1_	D_3_	D_4_	D_10_	D_9_	D_12_	D_22_	D_18_	D_20_
	Obstacle degrees	17.96%	16.95%	14.44%	15.60%	13.47%	12.08%	20.46%	19.87%	18.06%
2018	Obstacle factors	D_4_	D_2_	D_5_	D_12_	D_10_	D_9_	D_20_	D_21_	D_22_
	Obstacle degrees	16.98%	15.31%	13.70%	15.09%	14.33%	12.50%	21.72%	18.36%	17.87%
2019	Obstacle factors	D_2_	D_4_	D_1_	D_11_	D_9_	D_13_	D_22_	D_20_	D_21_
	Obstacle degrees	16.86%	16.05%	15.39%	15.12%	14.07%	13.74%	20.34%	17.12%	16.69%

## Data Availability

Restrictions apply to the availability of these data. Data was obtained from [National Bureau of Statistics of China] and are available [at https://data.stats.gov.cn/] with the permission of [National Bureau of Statistics of China].

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
