# Peer review of "Evolution and Influencing Factors of Social-Ecological System Vulnerability in the Wuling Mountains Area"

_ijerph, 2022, doi:10.3390/ijerph191811688_

Round 1
Reviewer 1 Report (Previous Reviewer 3)
Dear Authors,
Thank you for being so kind to take into consideration my comments and suggestions for your work! In my opinion, the work was improved substantially and may be accepted for publication as it is in the current form. Congratulations for your manuscript and I really appreciate your effort!
Good luck!
Kind regards,
The Reviewer
Author Response
What a delightful comment! We are very honored by your recognition. Thank you very much for your support and help in our work during this time period. Your comments and suggestions has brought about a great improvement to our paper. Thank you again for your great patience and huge help.
Reviewer 2 Report (New Reviewer)
The paper is a reasonably standard application of a vulnerability index. As a local study it is fine but there are two aspects that need further attention
1) how does the index compare to other similar exercises - where are their similarities and differences - and what are the implications for comparative analyses
2) please provide a fuller explanation as to why indices were selected and their appropriateness
Author Response
We really appreciate your efforts in reviewing our manuscript. Your careful review has helped to make our study clearer and more comprehensive. According to your suggestion, we compared this paper with similar studies in the selection of indicators, and also explained the basis for the selection of some special indicators in the first paragraph of 2.2. Construction of the Evaluation Index System.
Round 2
Reviewer 2 Report (New Reviewer)
Thank you for your efforts with revising the paper. It has improved. However, at times there needs to be a little bit more clarity that your additions relate to your specific study sight rather than being general comments.
it would still be appropriate to add a further additional comment or two with respect to the comparability of your results with other studies of vulnerability and the implications for policy making.
Author Response
Your comment is very significant, and it has inspired our research thinking. According to your suggestion, we supplemented some comparison of our results with those of similar studies. For these supplements, we think it is more appropriate to put them in the 4.3. Analysis of comparisons with similar studies, and if you have better ideas, we will be very glad. Thank you very much!
This manuscript is a resubmission of an earlier submission. The following is a list of the peer review reports and author responses from that submission.
Round 1
Reviewer 1 Report
This paper makes an assessment of the social-ecosystem vulnerability in the period 2010-2019, in 42 counties of the Wuling Mountain area, in which tourism constitutes an external resource that contributes to poverty alleviation. Precisely, the main contribution of this work is the dynamic analysis of social-ecosystem vulnerability, in relation to existing studies, also pointing out the lack of the large sample data in them.
In general terms, the work has an adequate quality, carrying out an in-depth analysis of the changes that have occurred in the characteristics of social-ecosystem vulnerability of the Wuling mountain area, and of the factors with the greatest incidence in said vulnerability.
However, in my opinion, I think this work can be improved by answering the following questions:
1. In the introduction, it would be necessary to define, more precisely, the objectives of the work, the questions that the research intends to answer, the scientific contribution that it makes with respect to the existing literature and the expected policy implications.
2. The "Vulnerability evaluation index system" in Table 1 is a synthetic indicator, so it is very important to further clarify how the weights of the variables have been assigned. It would be useful for the paper to answer the following questions:
a. What are the advantages of using the entropy method for assigning weights? Has it been compared to other methods of constructing synthetic indicators, such as principal component analysis, factor analysis, the P2 distance method or DEA?
b. On the other hand, the variables or partial indicators used are expressed in different units of measurement. Could you indicate whether there have been any problems of aggregation of these variables and how they have been solved?
c. Finally, have you detected any problems of multicollinearity between the variables used? Has any method been used to eliminate the redundant information provided by the variables when they are integrated into the synthetic indicator?
3. Table 4 provides an analysis of the vulnerability barrier factors of each subsystem". I would also find it interesting to know how the importance or weight of each of the three subsystems as a whole varies over the period 2010-2019, not only in the top three obstacle factors or variables. According to the data in Table 4, I think that some more factors can be highlighted that have some importance, such as D1 in the economic subsystem, D10 in the social subsystem and D21 in the ecological subsystem. In addition, the factors that have been gaining or losing weight over the period could be analyzed, as they can be used to detect trends that facilitate the design of more effective policies to reduce the vulnerability of the system.
4. Delve into the causes that explain the evolution of the economic, social and ecological subsystems in the period of analysis, in each of the four provinces, especially in the counties that have managed to further reduce the vulnerability of the system between 2010 and 2019. Have “successful” policies or measures been carried out in these counties, which explain the favorable evolution of social-ecosystem vulnerability?
5. Within the policy implications, it is essential to establish a closer relationship between the vulnerability factors detected in each area, in accordance with the trend they have followed over the period 2010-2019, and the measures proposed by the authors of the article to reduce the social-ecosystem vulnerability. It would be convenient to differentiate the measures that should be promoted in the short term from those that could have a longer-term implementation horizon.
I hope these suggestions can help you to further refine the work.
Author Response
Thank you for your commnets!Please see the attachment for the specific reply.

Reviewer 2 Report
Why is there seemingly a lot of change in the results from 2018 to 2019? I would think there wouldn't be much since it is only 1 year, but your results suggest otherwise. This should be explained.
This same line of explanation should also occur for each of the 2 year timeframes. What happened that these changes occurred and why?
The targeted poverty alleviation program is not the answer as there is inevitably a lag with policy.
The conclusions section needs to be rewritten to make more sense; especially the last paragraph.
Author Response
Thank you for your comments. PLease see the attachment for the specific reply.

Reviewer 3 Report
Dear Authors,
Please find below and attached my comments and suggestions for your work.
Good luck!
Kind regards,
The Reviewer
Review Report Form
Journal: IJERPH (ISSN 1660-4601)
Manuscript ID: ijerph-1829868
Type: Article
Title: Evolution and influencing factors of Social-Ecological System Vulnerability in Wuling Mountain Area
Authors: Huiqin Li * , Yujie Hui , Xin Cheng , Jingyan Pan
Section: Environmental Earth Science and Medical Geology
Special Issue: Impacts of Human Activities and Climate Change on Landscape
Submission Date: 07 July 2022
Dear Authors,
I have carefully analyzed your article entitled “Evolution and influencing factors of Social-Ecological System Vulnerability in Wuling Mountain Area”.
Congratulations for your work and valuable insights reflected in the content of the manuscript!
The structure of my Review Report Form takes into consideration two sections, namely: (A.) General overview of the article and strong points; and (B) Suggestions meant to improve your current manuscript.
(A.) General overview of the article and strong points:
Ø General background of the study & Aim of the study: The authors have mentioned that the wide spread of the concept of sustainable tourism in various countries and regions has led to the need to extend the research on tourism poverty alleviation, especially by focusing the attention to the sustainability of the poverty reduction effect of tourism, and the social-ecosystem theory of tourist destinations, in particular, with reference to the sustainable development of tourism in poor mountainous areas. Based on the authors’ notes, the authors believe that the existing studies lack the dynamic evaluation of social-ecosystem vulnerability in tourism places, and the large sample data are lacking.
Ø Research methodology used: The authors have pointed out that in order to accomplish the purpose of their work they have selected forty-two national key poverty-alleviation counties in Wuling Mountain area as the research objects to analyze the spatial and temporal evolution characteristics of social-ecosystem vulnerability, using in terms of research methodology the valuation model of "Social-Economic-Ecological (S-E-E) model" and the "Vulnerability-Scoping-Diagram (V-S-D) model".
Ø Results of the study: In terms of the results of this current study, it ought to be mentioned that that the authors found that: (a) from 2010 to 2019, the overall vulnerability of social-ecosystem showed a trend of "slow-rise and steady-decline", with the vulnerability index peaked in 2014 and declining year by year since then; (b) spatially, the overall vulnerability is superior in the north than in the south; and (c) social-ecosystem vulnerability is the result of the interaction between system exposure-sensitivity and system adaptive capacity.
(B) Suggestions meant to improve your current manuscript:
Distinguished Authors I would kindly like to suggest the following aspects:
(1.) Closely analyzing the article, since there are some English language improvements and slight corrections that need to be taken care of. Thus, my recommendation would be to carefully proofread the entire manuscript.
(2.) Also, I have closely analyzed the format of the article, in order to check whether it follows the guidelines which are specific to the publisher. Thus, I have noticed that the current form of your work needs improvement in this regard. So, my kind suggestion is to closely analyze again the guidelines belonging to the publisher, since the article should fit exactly the publisher’s guidelines. For instance, the keywords, the subsections, the references, currently do not fit the style and the requirements of the publisher. Also, it would be highly recommendable to include in the abstract of your study more highly relevant details that refer to the research objectives and the methodology used. This would definitely be considered a plus for your scientific work.
(3.) In continuation, the suggestion would also be inserting in your article a few ideas concerning the correlation between effects of the COVID-19 pandemic and the COVID-19 global crisis, sustainability and sustainability assessment, Sustainable Development Goals, while focusing on the evolution and influencing factors of social-ecological system vulnerability in Wuling Mountain Area, since these are key focuses these days. In this context, I had the chance to read a few interesting scientific works recently, among which I would like to mention: An Exploratory Study Based on a Questionnaire Concerning Green and Sustainable Finance, Corporate Social Responsibility, and Performance: Evidence from the Romanian Business Environment. J. Risk Financial Manag. 2019, 12, 162. https://doi.org/10.3390/jrfm12040162; OECD. Measuring the Impacts of Business on Well-Being and Sustainability. https://www.oecd.org/statistics/Measuring-impacts-of-business-on-well-being.pdf; OECD. 2022. Toward sustainable economic development through promoting and enabling responsible business conduct. https://www.oecd-ilibrary.org/sites/f7813858-en/index.html?itemId=/content/component/f7813858-en.
Dear Authors, congratulations once again for your work and valuable insights reflected in the content of the manuscript, and I hope my comments will be of value to you!
Kind regards,
The Reviewer
Author Response
Thank you for your comments. Please see the attachment for the specific reply.

Round 2
Reviewer 1 Report
Having analyzed the revised version of the article, and the authors' responses to my concerns, I believe that this work, although it can still be improved in some aspects, can be published in IJERPH. As the authors have commented, it would be necessary for them to make further modifications to the work in the future, in order to address some of the issues that have not been resolved in the current revision.
Author Response
Thank you very much for your recognition of our work. We must acknowledge that your comments has been very helpful to our paper, and we will further refine our paper in our future work.
Reviewer 2 Report
The revised paper is better.
Lines 894-895 Also lines 710-712: Improving the urbanization rate can improve the local tourism reception capacity, increase its tourist carrying capacity, and ease the conflicts and contradictions between the host and guests.
This is just wrong. Increasing the urbanization rate could put the whole system into decline. This statement should be removed.
Line 900: What is artificial deforestation?
Author Response
Point: Lines 894-895 Also lines 710-712: Improving the urbanization rate can improve the local tourism reception capacity, increase its tourist carrying capacity, and ease the conflicts and contradictions between the host and guests.
This is just wrong. Increasing the urbanization rate could put the whole system into decline. This statement should be removed.
Response: I am very sorry that I made such a mistake. I looked closely at Table 1 and I found the Urbanization rate (D11) belongs to Exposure-sensitivity, so it is positively correlated with system vulnerability. As you said, Increasing the urbanization rate could put the whole system into decline. So I removed the statement .
Point: Line 900: What is artificial deforestation?
Response: Regarding the "artificial deforestation" you mentioned, I did not find
both words in the paper. I only found "artificial afforestation". So I want to ask if you have a spelling error. If you mean "artificial afforestation," then it means artificially planting some trees and eventually forming a forest
I said "I did not accurately understand what Reviewer2 meant", which is mainly due to the following comment. This comment exists in reviewer2’s Review Report (Round 1).
Point1: Why is there seemingly a lot of change in the results from 2018 to 2019? I would think there wouldn't be much since it is only 1 year, but your results suggest otherwise. This should be explained. This same line of explanation should also occur for each of the 2 year time frames. What happened that these changes occurred and why?The targeted poverty alleviation program is not the answer as there is inevitably a lag with policy.
Response1: Thank you very much for your valuable advice, but we have some doubts that we hope you can answer. In your opinion, I have looked closely at 3.2. Spatial Evolution, and we have found that the vulnerability change of the three subsystems from 2018 to 2019 is not very large, which can be clearly seen from Figures 3,4, and 5. In Figure 2 of 3.1. Timing Variation, we also found that the changes from 2018-2019 were also not obvious in the 2-year time frame. Maybe I don't understand what you mean, what do you mean by the change? If you can, I hope you can point it out in detail, thank you very much for your hard work.
Thank you very much for your patient and careful work!